# Effects of COVID-19 Lockdown on Heart Failure Patients: A Quasi-Experimental Study

**DOI:** 10.3390/jcm12227090

**Published:** 2023-11-14

**Authors:** Juan Luis Sánchez-González, Luis Almenar-Bonet, Noemí Moreno-Segura, Francisco Gurdiel-Álvarez, Hady Atef, Amalia Sillero-Sillero, Raquel López-Vilella, Iván Santolalla-Arnedo, Raúl Juárez-Vela, Clara Isabel Tejada-Garrido, Elena Marques-Sule

**Affiliations:** 1Department of Nursing and Physiotherapy, University of Salamanca, 37007 Salamanca, Spain; juanluissanchez@usal.es; 2Heart Failure and Transplantation Unit, Department of Cardiology, La Fe University and Polytechnic Hospital, 46026 Valencia, Spain; lualmenar@gmail.com (L.A.-B.); lopez_raqvil@gva.es (R.L.-V.); 3Centro de Investigación Biomédica en Red de Enfermedades Cardiovasculares (CIBERCV), Instituto de Salud Carlos III, 28933 Madrid, Spain; 4Department of Medicine, Universidad de Valencia, 46010 Valencia, Spain; 5Department of Physiotherapy, Faculty of Physiotherapy, University of Valencia, 46010 Valencia, Spain; noemi.moreno@uv.es; 6Escuela Internacional de Doctorado, Department of Physical Therapy, Occupational Therapy, Reha-Bilitation and Physical Medicine, Universidad Rey Juan Carlos, 28933 Alcorcón, Spain; 7School of Allied Health Professions (SAHP), Keele University, Keele, Staffordshire ST5 5BG, UK; hady612@hotmail.com; 8University School of Nursing and Physiotherapy “Gimbernat”, Autonomous University of Barcelona, Avd de la Generalitat, 202-206, Sant Cugat del Vallès, 08174 Barcelona, Spain; amaliasillero@hotmail.com; 9ESIMar (Mar Nursing School), Parc de Salut Mar, Universitat Pompeu Fabra Affiliated, 08018 Barcelona, Spain; 10SDHEd (Social Determinants and Health Education Research Group), IMIM (Hospital del Mar Medical Research Institute), 08003 Barcelona, Spain; 11Nursing Department, Faculty of Health Sciences, Research Group GRUPAC, University of La Rioja, 26004 Logroño, Spain; raul.juarez@unirioja.es (R.J.-V.); clara-isabel.tejada@unirioja.es (C.I.T.-G.); 12Physiotherapy in Motion, Multispeciality Research Group (PTinMOTION), Department of Physiotherapy, Faculty of Physiotherapy, University of Valencia, 46010 Valencia, Spain

**Keywords:** COVID-19, heart failure, physical activity, quality of life, sleep quality

## Abstract

Introduction: The COVID-19 lockdown has been associated with reduced levels of physical activity, quality of life, and sleep quality, but limited evidence exists for its impact on heart failure patients. This study examined the influence of the COVID-19 lockdown on these aspects in heart failure patients, with specific comparisons by age and sex. Methods: A quasi-experimental cross-sectional study of patients with heart failure was conducted. The assessment involved two time points: during the COVID-19 lockdown (March to June 2020) and post-lockdown (July to October 2020). A total of 107 HF patients participated, with assessments of overall PA (using the International Physical Activity Questionnaire), QoL (employing the Cantril Ladder of Life), and sleep quality (utilizing the Minimal Insomnia Symptom Scale) conducted during and after the COVID-19 lockdown. Results: HF patients reported lower levels of total PA (*p* = 0.001) and walking PA (*p* < 0.0001) during lockdown than after lockdown, whilst no differences were observed in QoL nor sleep quality. In addition, both younger and older patients reported lower walking PA and total PA during lockdown than after lockdown, while older patients reported lower QoL during lockdown than after lockdown. Moreover, both men and women reported lower walking PA and total PA during lockdown than after lockdown, whilst women reported lower QoL. Conclusions: HF patients need improved PA programs during lockdowns, as these programs can elevate PA levels and enhance QoL, especially when faced with the risk of decompensation during health crises.

## 1. Introduction

At the beginning of the pandemic in 2020, many countries opted to contain the spread of COVID-19 by shutting down most significant activities to reduce the pressure on the national health system and avoid increasing deaths [1]. In Spain, the first case of COVID-19 was reported on 1 January 2020, and on 14 March 2020, the government approved a nationwide lockdown. That action prohibited wandering in public spaces except under exceptional circumstances from March to June 2020. These lockdown measures interfered with the general population’s daily life and physical and psychological health [2,3], including health professionals and university students [4]. Furthermore, previous studies have demonstrated that this situation has had several consequences on the cardiometabolic system, morbidity, and mortality levels, even in healthy subjects [5]. The situation was even worse with older adults [6] and patients with cardiovascular diseases such as heart failure (HF) [7]. However, Spain was not the only country affected; Asia, Europe, and America were also affected both by the infection itself and by the measures adopted [8,9].

Heart failure HF is a lifelong condition in which the heart cannot pump enough blood to meet the body’s needs for blood and oxygen [10]. The same happened to all people during the COVID-19 lockdown, and HF patients had similarly experienced a reduction of physical activity and lockdown-related psychological affection [2,11]. Moreover, this population is more vulnerable to suffering severe COVID-19 and its complications. A recent review by Harrison et al. [12] indicated that cardiovascular disease, specifically HF, is associated with an increased risk of severe COVID-19 and mortality from COVID-19 [12]. This higher proportion of adverse effects could be explained by underlying changes (such as lifestyle), which affect inflammatory pathways and immune and pulmonary functions [12].

The primary variable affected during the pandemic was the physical activity (PA) performed due to the inability to walk regularly outside during the prolonged lockdown [5]. This sedentary behavior can also increase the risk of suffering health problems such as diabetes [13], cancer [14], or osteoporosis [15]. Considering that patients with cardiovascular diseases such as HF have shown lower values of PA than healthy adults [16] may explain why these patients’ clinical condition may have been aggravated during the mandatory COVID-19 lockdowns [17,18]. Moreover, in the study performed by Vetrovsky et al. [2], the number of daily steps in 26 patients with HF was analyzed for six weeks, and three of these weeks were during the COVID-19 lockdown period. They found a significant decrease of 1134 daily steps during the lockdown. In parallel, the study of Brasca et al. [19] showed a reduction of 6.5% in PA levels during the COVID-19 lockdown in a cohort of 405 HF patients. Previous studies have demonstrated that the reduction of PA levels tended to be more significant and maintained post-lockdown in those patients who were less physically active before the COVID-19 lockdown and had a more significant NYHA classification [19,20]. The study of Bakel et al. [17] explained that the lack of social contact limited the possibilities of performing PA, whilst younger age was independently associated with a higher sedentary behavior during lockdown.

The consequences of this reduction of the PA levels in HF patients went beyond physical and cardiometabolic sequelae. Previous studies have explained that decreased levels of PA could also affect mental health and quality of life [21,22,23]. For example, the review of Brooks et al. [3] showed that those who underwent forced lockdown had an increased risk of experiencing some episode of stress, decreasing their emotional quality of life. Sang et al. [22] showed that the COVID-19 lockdown created various psychological impacts, negatively impacting the emotional status due to depression and anxiety. Other studies showed an increased incidence of sleep disturbances in the general population during the first months of the pandemic [16,17]. For example, Voitsidis et al. [24] showed that 37.6% of the Greek general population presented alterations in sleep quality.

Furthermore, the literature has strong evidence that sleep quality disturbance has been associated with an increased risk of cardiovascular events [25]. In addition, people with chronic illnesses such as HF have an increased risk of mood and sleep disturbances that affect their quality of life [26,27,28]; thus, a more in-depth study of these aspects would be necessary. In consistency with these studies, Quintana et al. [29] evaluated the influence of the COVID-19 lockdown on the quality of life in HF patients. They concluded that during the COVID-19 lockdown, participants showed reduced ability to enjoy daily activities and self-confidence and decreased quality of life.

However, the population with HF is very heterogeneous regarding sociodemographic characteristics. Previous studies on the general population during the COVID-19 lockdown have been performed in this regard. Faulkner et al. [30] assessed physical activity, mental well-being, and quality of life among adults in the United Kingdom, Ireland, New Zealand, and Australia during the COVID-19 lockdown. Their findings revealed that women experienced more favorable physical activity and mental health improvements than men. Additionally, younger participants reported more adverse changes in these aspects than their older counterparts. Beck et al. [31] showed, in a cohort of more than 1000 participants, that approximately 75% of the subjects reporting sleep quality problems were women. Young participants were the ones who presented more sleep issues. Thus, it is evident that the pandemic has affected such outcomes differently depending on age and sex.

Therefore, there is enough evidence to report the changes in PA, quality of life, and sleep quality in the general population during the COVID-19 lockdown. The evidence on HF patients is scarce, and, to the best of our knowledge, no literature makes a differentiation based on sex and age in HF patients in this regard; therefore, this research is novel in this field. The present study explored the PA, quality of life, and sleep quality in HF patients during and after the COVID-19 lockdown and compared the results by sex and age.

The hypotheses of the study are as follows:

**H1:** 
*The physical activity in HF patients during the COVID-19 lockdown was lower than after the lockdown.*


**H2:** 
*The quality of life in HF patients during the COVID-19 lockdown was lower than after the lockdown.*


**H3:** 
*The sleep quality in HF patients during the COVID-19 lockdown was lower than after the lockdown.*


**H4:** 
*Physical activity, quality of life, and sleep quality are different when comparing women with HF with men during the COVID-19 lockdown.*


**H5:** 
*Physical activity, quality of life, and sleep quality are different when comparing those aged ≥ 65 years old with those aged < 65 years old.*


## 2. Materials and Methods

### 2.1. Design and Setting

This study employed a cross-sectional, pre-post, quasi-experimental design without a control group. Patient recruitment occurred from March 2020 to October 2020 in Valencia, Spain. Written informed consent was obtained from all participants.

### 2.2. Sample

This study was completed with a total of 107 participants with HF at different outpatient clinics in Valencia, Spain. Inclusion criteria were as follows: (1) participants above 18 years of age (2) who have a clinical diagnosis of HF and (3) who are cognitively capable of completing the assessments. Participants with cognitive and neuropsychiatric disorders were excluded from this study.

### 2.3. Procedure

Participants were clinically diagnosed with heart failure (HF) based on electronic medical records. The assessment involved two time points: during the COVID-19 lockdown (March to June 2020) and post-lockdown (July to October 2020), as depicted in Figure 1.

Due to the unique circumstances posed by the COVID-19 pandemic, our data collection strategy was adapted. Despite the initial recruitment being carried out in a face-to-face setting, we opted for telephone interviews to ensure the safety and well-being of our participants. These interviews were conducted by a trained researcher who adhered to a structured and standardized approach.

### 2.4. Outcomes and Measures

A patient information form was developed to collect data regarding demographic and clinical variables: Our study encompassed a range of sociodemographic and clinical variables, including gender, age, marital status, working status, education, time since diagnosis, weight, height, and body mass index (BMI). Additionally, we assessed the following variables:

(1) Overall physical activity: This was measured with the Spanish version of the International Physical Activity Questionnaire (IPAQ) [32]. The IPAQ contains seven questions to determine the duration and frequency of light activity (<600 metabolic minutes: walking at home and at work or any walking that can be done solely for leisure); moderate activity (between 600 and 3000 MET minutes/week such as cycling, playing tennis, or carrying light loads); and vigorous activity (>3000 MET minutes/week: doing aerobic exercise, digging, or heavy lifting). It also assesses the inactivity of the last week. The total PA score is the sum of vigorous, moderate, and light PA in MET minutes/week [32,33]. The questionnaire presents a total intra-observer reliability of 0.914 and 0.900 in the three dimensions of the questionnaire separately. Moreover, the questionnaire presents an internal consistency of Cronbach’s alpha = 0.51 [34].

(2) Quality of life (QoL): QoL was assessed using the Cantril Ladder of Life [35]. This questionnaire has been employed in previous cardiovascular studies and is considered a valid measure of global quality of life [36,37,38]. It does not cover quality of life as a multidimensional concept, although it is related to aspects of quality of life such as psychosocial adjustment and functional capabilities [39]. Patients were asked about their quality of life on a scale from 0 to 10 (scores of 10 reflected the best quality of life and 0 reflected the worst). The scale shows good convergent validity [40] and presents a total intra-observer reliability of 0.914 [40,41]. This scale was translated to Spanish by a native Spanish speaker and was back-translated by another independent bilingual researcher.

(3) Quality of sleep: Sleep quality was assessed by the Minimal Insomnia Symptom Scale (MISS) [42,43], which consists of three items with five response categories (no, minor, moderate, severe, and very severe problems) that are scored from 0 to 4, respectively. The total score ranged from 0 to 12, with higher scores representing higher sleeping difficulties. The questionnaire asks about difficulties falling asleep, nighttime awakenings, and rest during sleep. The reliability and validity of the MISS have been established among the elderly with a high intraclass correlation coefficient of 0.79 with an internal consistency of Cronbach’s alpha = 0.73 [43]. This questionnaire was translated to Spanish by a native Spanish speaker and was back-translated by another independent bilingual researcher.

Reliability and validity of the questionnaires shows in Table 1.

### 2.5. Data Analysis

Statistical analyses were performed using the statistical package SPSS version 24 (IBM SPSS, Inc., Chicago, IL, USA). Mean, standard deviation, and percentage were used to describe the sample data. Sample size calculation: We determined the required sample size before the study. With an estimated alpha risk of 0.05 and a beta risk of less than 0.2 in a bilateral contrast, we aimed to have sufficient statistical power to detect differences of 15% or more. Based on these criteria and assuming a standard deviation of 60 for the difference between measurements taken before and after the lockdown, we determined that a sample size of 100 subjects was necessary for our analysis. This ensured that the study would have the statistical strength to identify significant changes in the variables of interest. The Kolmogorov–Smirnov test was used to verify the normality of the continuous data. Paired t-tests for the Cantril Ladder of Life, MISS, and IPAQ were employed to compare the differences between COVID-19 lockdown and after-lockdown periods within matched pairs. The effect size was calculated using Cohen’s d for the paired *t*-test. Subgroup analyses were also performed for age (<65 years and ≥65 years) and sex. For the subgroup analysis, paired t-tests were used to compare between time events and Student *t*-tests were used to compare between groups. Statistical significance was considered when *p*-values were <0.05.

### 2.6. Ethical Considerations

Approval was obtained from the Regional Ethics Committee (Approval No. IE1529273). All procedures were conducted strictly within the principles of the Declaration of Helsinki (October 2013, Fortaleza, Brazil). The researchers explained the research aims to all participants. Stringent measures were taken to ensure the privacy and confidentiality of participant data. The anonymity of the participants was guaranteed, and informed consent to participate was obtained prior to data collection. These ethical measures were implemented to safeguard the rights and well-being of the study participants.

## 3. Results

A total of 107 HF patients were assessed during and after the COVID-19 lockdown. The characteristics of the 107 participants are shown in Table 2.

Regarding PA, HF patients reported significantly lower levels of walking PA during lockdown than after lockdown (r = 0.49, *p* < 0.001). In addition, significantly lower levels of total PA were also found when comparing during lockdown and after lockdown periods (r = 0.46, *p* < 0.001). Nevertheless, no differences were found in vigorous PA (*p* = 0.181) or moderate PA (*p* = 0.068) levels or in sedentary time (*p* = 0.872). With regards to quality of life, no differences were found during lockdown compared to after lockdown (*p* = 0.091). Regarding sleep quality, no differences were found during lockdown compared to after lockdown (difficulties falling asleep, *p* = 0.897; night awakenings, *p* = 1.000; not being rested by sleep, *p* = 0.495) (Table 3).

When comparing by age, both those aged <65 years old and those aged ≥65 years old reported significantly lower levels during a lockdown than after lockdown in walking PA (<65 years: r = 0.52, *p* = 0.001; ≥65 years: r = −0.48, *p* < 0.001), as well as in total PA (<65 years: r = 0.52, *p* = 0.003; ≥65 years: r = 0.45, *p* < 0.001), whilst there were no differences in the rest of the IPAQ variables. In addition, when comparing age subgroups (<65 years old vs. ≥65 years) during lockdown, we did not find significant differences in any of the IPAQ variables (vigorous PA (*p* = 1.000), moderate PA (*p* = 0.451), walking PA (*p* = 0.256), sedentary time (*p* = 0.264), total PA (*p* = 0.203)). When comparing the quality of life by age, those aged ≥ 65 years old reported significantly lower quality of life (r = 0.15, *p* = 0.039) during lockdown than after lockdown. When comparing sleep quality by age, we did not find differences between groups or intra-groups (Table 4).

When comparing by sex, both men and women reported significantly lower levels during lockdown than after lockdown in walking PA (men: r = 0.57, *p* < 0.001; women: r = 0.47, *p* < 0.001), as well as in total PA (men: r = 0.54, *p* < 0.001; women: r = 0.07, *p* < 0.001), while there were no differences in the rest of the IPAQ variables. In addition, when comparing by sex during lockdown, we did not find significant differences in vigorous PA (*p* = 1.000), moderate PA (*p* = 0.289), walking PA (*p* = 0.069), sedentary time (*p* = 0.071), or total PA (*p* = 0.135). When comparing the quality of life by sex, women reported a significantly lower quality of life (r = 0.21, *p* = 0.046) during lockdown than after lockdown. When comparing sleep quality by sex, we did not find any difference between women and men (Table 5).

## 4. Discussion

In this study, we investigated the physical activity (PA), quality of life, and sleep quality in heart failure (HF) patients during and after the COVID-19 lockdown, with a focus on differences related to age and sex.

### 4.1. Physical Activity (PA) during Lockdown

Our study findings reveal that HF patients, both men and women, irrespective of age (above or below 65 years), experienced a decline in their PA levels during the COVID-19 lockdown. However, once the lockdown restrictions were lifted, there was an increase in PA. Notably, participants exhibited higher sedentary behavior during the lockdown, validating our initial hypothesis (H1: The PA in HF patients during the COVID-19 lockdown was lower than after the lockdown). These results are consistent with previous research, such as that of Tison et al. [44], who noted an increase in daily steps post-lockdown, and Van Bakkel et al. [18], who observed progressively increasing sedentary behavior as restrictions eased. Nevertheless, a recent systematic review emphasized a significant decrease in PA levels among patients with cardiovascular diseases, particularly HF patients, during the COVID-19 lockdown [45]. Subgroup analysis indicated that the decrease in PA was consistent across gender and age, leading us to reject hypotheses H4 (differences in PA between women and men with HF during the lockdown) and H5 (differences in PA between those aged ≥ 65 and <65 years). Kim et al. [46] found different results in the sex difference in a sample of 229,099 subjects. They observed that men engaged in more moderate-intensity PA than women before and during COVID-19. On the other hand, Punia et al. [47] observed that men showed lower physical activity levels during the pandemic in a sample of 1992 subjects. Therefore, there seems to be too much discrepancy between BP levels between the two genders.

### 4.2. Quality of Life during Lockdown

Contrary to our expectations (H2), the overall quality of life among HF patients did not exhibit any significant changes during the COVID-19 lockdown in our study. However, variations were observed when examining different age and gender groups, leading us to accept hypotheses H4 (differences in quality of life between women and men with HF during the lockdown) and H5 (differences in quality of life between those aged ≥65 and <65 years). Specifically, adults older than 65 and women reported a lower quality of life during the lockdown than those younger than 65 years and men. It is important to note that an HF diagnosis can substantially impact a patient’s quality of life [29]. These results differ from a recent epidemiological study [29], which reported that patients with HF had difficulties enjoying daily activities during the COVID-19 lockdown, although no post-lockdown assessments were conducted. As such, it is crucial to emphasize self-care behaviors and provide practical self-care management information to HF patients and their families. To our knowledge, this is the first study that reports the results about the quality of life of HF patients during lockdown, but previous studies were performed on other health problems such as the study of van Erck et al. [48], who found an increase in the quality of life during lockdown in patients awaiting transcatheter aortic valve implantation. In contrast, Banerjee et al. [49] showed that the quality of life of patients with Parkinson’s Disease and their caregivers was decreased. This lack of differences could be due to the fact that patients with HF have a chronic clinical condition that limits their quality of life; therefore, the modifications in quality of life due to confinement could be more subtle. Then, future studies that explored the changes in quality of life in diseases that already had a reduced quality of life should be performed in order to understand these results better.

### 4.3. Sleep Quality during Lockdown

Our study showed no discernible differences in sleep quality between different periods, genders, or age groups. As a result, hypotheses H3 (that sleep quality in HF patients during the COVID-19 lockdown was lower than after lockdown), H4 (that sleep quality differed between women and men with HF during the COVID-19 lockdown), and H5 (that sleep quality varied between those aged ≥ 65 years and <65 years) were not supported. However, various studies have highlighted the severe impact of the COVID-19 pandemic on multiple aspects of human life and its substantial threat to the mental and physical health of the general population [50]. Our results do not align with a study by Okely et al. [51], who investigated changes in sleep quality among elderly individuals during the COVID-19 lockdown. They also explored whether participant characteristics were related to positive or negative changes in sleep quality, ultimately concluding that participants with a history of cardiovascular disease had worse sleep quality during the lockdown. As with the quality of life, to our knowledge, this is the first study that analyzes the quality of sleep of HF patients during lockdown, but studies performed in patients with pulmonary hypertension [52], women with polycystic ovary syndrome [53], and also in the general population [54] showed a reduced quality of sleep during lockdown [52]. A reason for these results could be the fact that the HF patients had a reduced PA in normal conditions; therefore, although a reduction in PA was evidenced in the lockdown period, it could be insufficient to reduce the quality of sleep.

Study limitations: Our study had several limitations to consider. First, the relatively small sample size and the predominance of male participants introduced potential gender bias, limiting the generalizability of findings to a broader heart failure (HF) population. Second, the exclusive focus on HF patients within a single country restricted the generalizability of results across diverse international contexts. Additionally, the reliance on self-reported questionnaires for assessing physical activity (PA) may have introduced recall bias and overestimation. We recognize the need for validity and reliability testing on the questionnaires to enhance the study’s robustness. Furthermore, the absence of data quality checks may imply potential data quality issues. It is also worth noting that socioeconomic status and mental health are possible factors that could influence the PA levels in this population, which, in conjunction with the study design, account for a causal inference problem in the results. Nonetheless, future investigations can explore data accuracy, completeness, reliability, relevance, and timeliness.

Strengths: Our study offered a comprehensive assessment of PA, quality of life, and sleep quality changes in HF patients during and after the COVID-19 lockdown, which were areas with limited prior research. Conducting measurements at two time points provided valuable insights into the impact of the pandemic and lockdown measures on these aspects of patients’ lives. In summary, despite these limitations, our study contributes significant insights into the experiences of HF patients during and after the COVID-19 lockdown, and future research should aim to address these constraints and further enrich our understanding of this critical issue.

### 4.4. Implications and Future Lines of Action

Physical activity (PA) constitutes a pivotal intervention in heart failure (HF) management, with research underscoring its significant impact on patient survival rates, including reduced all-cause mortality and HF-related mortality [55,56]. Therefore, addressing the risk of HF decompensation due to physical inactivity remains a paramount concern, necessitating the encouragement of HF patients to uphold substantial PA levels and minimize prolonged sedentary behavior, especially in contexts like the COVID-19 lockdown. This underscores the imperative need for developing home-based PA programs and implementing routine follow-up assessments for HF patients, emphasizing monitoring their PA levels.

Furthermore, preserving the high quality of life and overall health in the HF population is essential for averting decompensation and fostering well-being. To this end, a compelling strategy involves the integration of eHealth initiatives, specializing in medical care and cardiac telerehabilitation, thus offering PA-centric programs and services tailored to HF patients [57,58,59]. These eHealth initiatives should closely align with the exercise guidelines set forth by the American Heart Association [60] because they emphasize activities like moderate aerobic exercise for a minimum of 150 min per week, bi-weekly muscle strengthening routines, regular stretching, and a medical consultation before commencing any exercise regimen. Notably, in 2019, the European Commission’s Digital Economy and Society Index (DESI) identified Spain as the third-ranking country within the European Union in eHealth utilization [61]. This statistical revelation reveals Spain’s prior commitment to eHealth solutions, even preceding the global pandemic, underscoring the compelling prospects that future research should explore in harnessing these digital tools to enhance the welfare of HF patients.

## 5. Conclusions

Our findings revealed that regardless of age, HF patients experienced reduced walking and total PA levels during the lockdown, while quality of life and sleep remained relatively stable. Age-wise, both younger and older patients showed decreased PA levels during lockdown, with older patients also reporting reduced quality of life during this period. Sex-based comparisons indicated that men and women experienced declines in PA, while women reported decreased quality of life. Sleep quality, however, exhibited no significant sex-based differences. These results highlight the need for innovative strategies to boost PA and promote healthier lifestyles, particularly during public health crises like the COVID-19 lockdown. Implementing eHealth services and telerehabilitation programs could be valuable solutions to support patients. Engaging HF patients in PA-based initiatives elevates their PA levels and enhances their overall quality of life. It is especially critical when reduced PA may heighten the risk of disease exacerbation and increased patient vulnerability, especially when healthcare resources are constrained. These policies should extend beyond the HF population to encompass the entire community, contributing to maintaining a healthier population and reducing the strain on healthcare resources during health system stress.

## Figures and Tables

**Figure 1 jcm-12-07090-f001:**
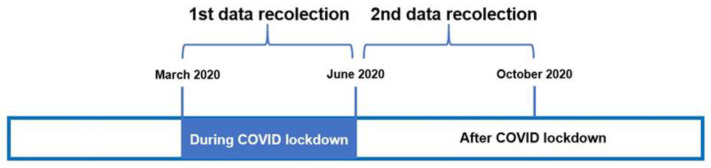
Timeline.

**Table 1 jcm-12-07090-t001:** Reliability and validity of the questionnaires.

	Intra-Observer Reliability	Internal Consistency
Physical activity (IPAQ)	0.900–0.914	0.51
Quality of life (Cantril Ladder of Life)	0.914	-
Quality of sleep (MISS)	0.79	0.73

**Table 2 jcm-12-07090-t002:** Clinical and demographic characteristics of the participants.

		By Sex		By Age	
Variables	Total (n = 107)	Male	Female	<65	≥65
Age, (years), mean (SD)	73.18 (12.68)	73.54 (12.91)	72.70 (12.51)	53.48 (8.78) *	77.99 (7.95) *
Sex, n (%)					
Male	61.00 (57.00)	61.00 (100.00)	0.00 (0.00)	12.00 (57.10)	49.00 (57.00)
Female	46.00 (43.00)	0.00 (0.00)	46.00 (100.00)	9.00 (42.90)	37.00 (43.00)
Marital status, n (%)					
Married	90.00 (84.10)	52.00 (85.20)	38.00 (82.60)	70.00 (81.40)	70.00 (81.40)
Single	2.00 (1.90)	2.00 (3.30)	0.00 (0.00)	1.00 (1.20)	1.00 (1.20)
Widowed	15.00 (14.00)	7.00 (11.5)	8.00 (17.40)	15.00 (17.40)	15.00 (17.40)
Working status, n (%)					
Employed	1.00 (10.20)	7.00 (11.40)	4.00 (8.70)	10.00 (47.70)	1.00 (1.20)
Unemployed	4.00 (3.70)	2.00 (3.30)	2.00 (4.30)	4.00 (19.00)	0.00 (0.00)
Housekeeper	14.00 (13.10)	5.00 (8.20)	9.00 (19.60)	0.00 (0.00)	14.00 (16.30)
Retired	78.00 (72.90)	47.00 (77.00)	31.00 (67.40)	7.00 (33.30)	71.00 (82.50)
Education, n (%)					
None	17.00 (15.80)	7.00 (11.40)	10.00 (21.70)	0.00 (0.00)	17.00 (19.80)
Primary education	39.00 (36.40)	24.00 (39.30)	15.00 (32.60)	2.00 (9.50)	37.00 (43.00)
Secondary education	38.00 (35.50)	20.00 (32.80)	13.00 (28.30)	14.00 (66.60)	24.00 (27.90)
University	13.00 (12.10)	8.00 (13.10)	5.00 (10.90)	5.00 (23.80)	8.00 (9.30)
Time since diagnosis, months, mean (SD)	96.54 (134.81)	82.75 (121.87)	114.83 (149.69)	64.38 (76.85)	104.40 (144.76)
LVEF, %, mean (SD)	43.36 (15.44)	40.51 (15.00) *	47.12 (15.38) *	41.74 (13.60)	43.75 (15.91)
NYHA Classification, n (%)					
I	7.00 (6.50)	5.00 (8.20)	2.00 (4.30)	1.00 (4.80)	6.00 (7.00)
II	73.00 (68.2)	43.00 (70.50)	30.00 (65.20)	15.00 (71.40)	58.00 (67.40)
III	22.00 (20.6)	10.00 (16.40)	12.00 (26.10)	5.00 (23.80)	17.00 (19.80)
IV	5.00 (4.70)	3.00 (4.90)	2.00 (4.30)	0.00 (0.00)	5.00 (5.80)
Weight, kilograms, mean (SD)	71.78 (14.11)	75.59 (13.80) *	66.72 (13.02) *	74.38 (14.64)	71.14 (13.99)
BMI, mean (SD)	26.42 (4.73)	26.51 (4.58)	26.30 (4.97)	27.45 (4.42)	26.17 (4.79)

Note: SD = standard deviation; BMI = body mass index; LVEF = left ventricular ejection fraction; NYHA = New York Heart Association. * *p* < 0.05.

**Table 3 jcm-12-07090-t003:** Results of physical activity, quality of life, and sleep quality during and after the COVID-19 lockdown.

	During Lockdown Mean (SD)	After Lockdown Mean (SD)	T Student *p* Value
Physical activity (IPAQ)			
Vigorous PA, METS minute/week	0.00 (0.00)	26.92 (206.74)	0.181
Moderate PA, METS minute/week	87.48 (329.90)	129.35 (424.27)	0.068
Walking, METS minute/week	302.61 (371.98)	871.23 (931.94)	<0.001 *
Sedentary time, hours/day	5.88 (5.41)	5.80 (4.66)	0.872
Total score, METS minute/week	386.85 (581.69)	999.16 (10)	<0.001 *
Quality of life (Cantril Ladder of Life)	5.61 (2.32)	5.84 (2.31)	0.091
Quality of sleep (MISS)			
Difficulties falling asleep	1.40 (1.46)	1.41 (1.45)	0.897
Night awakenings	1.51 (1.29)	1.51 (1.33)	1.000
Not being rested by sleep	0.63 (1.04)	0.59 (1.00)	0.495

Note: SD = standard deviation; IPAQ = International Physical Activity Questionnaire; MISS = Minimal Insomnia Symptom Scale; *: *p* < 0.05.

**Table 4 jcm-12-07090-t004:** Comparison of physical activity, quality of life, and sleep quality during and after the COVID-19 lockdown by age.

	<65 Years Old Mean (SD)		*p* Value	≥65 Years Old Mean (SD)		*p* Value
	During Lockdown	After Lockdown		During Lockdown	After Lockdown	
Physical activity (IPAQ)						
Vigorous PA, METS/min/week	0.00 (0.00)	91.43 (418.98)	0.329	0.00 (0.00)	11.16 (103.52)	0.320
Moderate PA, METS/min/week	80.00 (366.61)	80.00 (366.61)	1.000	89.30 (322.62)	141.40 (438.29)	0.068
Walking PA, METS/min/week	196.43 (244.12)	891.79 (899.75)	<0.001 *	328.85 (394.03)	866.15 (944.86)	<0.001 *
Sedentary time	7.10 (5.30)	7.38 (3.58)	0.788	5.58 (5.42)	5.42 (4.83)	0.756
Total score, METS/min/week	276.43 (488.48)	1063.21 (1180.22)	0.003 *	413.81 (601.73)	983.52 (1069.83)	<0.001 *
Quality of life (Cantril Ladder of Life)	6.33 (2.31)	6.24 (1.90)	0.776	5.43 (2.30)	5.74 (2.40)	0.039 *
Sleep quality (MISS)						
Difficulties falling asleep	1.33 (1.53)	1.48 (1.66)	0.526	1.42 (1.45)	1.40 (1.40)	0.748
Night awakenings	1.24 (1.22)	1.38 (1.28)	0.379	1.58 (1.31)	1.55 (1.34)	0.671
Not being rested by sleep	0.43 (0.81)	0.43 (0.81)	1.000	0.67 (1.09)	0.63 (1.04)	0.436

Note: Md = median; IQ range = interquartile range; SD = standard deviation; MISS = Minimal Insomnia Symptom Scale; IPAQ = International Physical Activity Questionnaire; *: *p* < 0.05.

**Table 5 jcm-12-07090-t005:** Comparison of physical activity, quality of life, and sleep quality during and after the COVID-19 lockdown by sex.

	Men Mean (SD)		*p* Value	Women Mean (SD)		*p* Value
	During Lockdown	After Lockdown		During Lockdown	After Lockdown	
Physical activity (IPAQ)						
Vigorous PA, METS/min/week	0.00 (0.00)	47.21 (273.01)	0.182	0.00 (0.00)	0.00 (0.00)	1.000
Moderate PA, METS/min/week	78.69 (324.14)	106.23 (343.70)	0.366	99.13 (340.64)	160 (514.60)	0.084
Walking PA, METS/min/week	347.33 (374.82)	929.14 (880.67)	<0.001 *	242 (393.44)	792.73 (1002.01)	<0.001 *
Sedentary time	6.72 (5.36)	6.57 (4.34)	0.815	4.76 (5.32)	4.78 (4.92)	0.975
Total score, METS/min/week	426.18 (527.38)	1082.58 (1060.86)	<0.001 *	334.70 (649.09)	888.54 (1123.06)	<0.001 *
Quality of life (Cantril Ladder of Life)	5.70 (2.25)	5.87 (2.22)	0.450	5.48 (2.43)	5.80 (2.46)	0.046 *
Sleep quality (MISS)						
Difficulties falling asleep	1.44 (1.50)	1.49 (1.49)	0.643	1.35 (1.41)	1.30 (1.40)	0.642
Night awakenings	1.57 (1.35)	1.56 (1.37)	0.874	1.43 (1.22)	1.46 (1.28)	0.830
Not being rested by sleep	0.61 (1.05)	0.61 (1.05)	0.471	0.59 (0.98)	0.57 (0.94)	0.811

Note: SD = standard deviation; MISS = Minimal Insomnia Symptom Scale; IPAQ = International Physical Activity Questionnaire; *: *p* < 0.05.

## Data Availability

Data will be provided upon request to the corresponding author.

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
