# Peer review of "Effects of COVID-19 Lockdown on Heart Failure Patients: A Quasi-Experimental Study"

_jcm, 2023, doi:10.3390/jcm12227090_

Round 1

Reviewer 1 Report

Comments and Suggestions for Authors

The authors evaluated the effects of of COVID-19 lockdown on heart failure (HF) patients by comparing PA, QoL and sleep quality during and after the COVID-19 lockdown. However, the major contribution to this topic is that it provides new data in HF patients. 

Comments:  This study investigated the effect of the COVID-19 lockdown on physical activity, quality of life, and sleep quality in heart failure (HF) patients, and further examined the difference by age and sex. The authors found lower levels of total or walking PA during lockdown both in younger and older patients. The paper wrote well, however, I have some comments as below.

1. The greatest strength is that this paper added new data about the effects of the COVI-19 lockdown on HF patients. However, the biggest problem is whether it supplied the sufficient novelty. The authors should better clarify the novelty, significance in the abstract and introduction.

2. The another strength of this study is to examine the difference with specific comparisons by age and sex, as a results, the authors should added the results of clinical and demographic characteristics stratified by age and sex.  

3. Except for lockdown, I think some other issues such as the social economic status, and mental health may also influence on the above aspects, and the causal inference problem should be noticed.   4.  The full stop is not necessary after the title.

Author Response

The authors evaluated the effects of of COVID-19 lockdown on heart failure (HF) patients by comparing PA, QoL and sleep quality during and after the COVID-19 lockdown. However, the major contribution to this topic is that it provides new data in HF patients. 

Comments:  This study investigated the effect of the COVID-19 lockdown on physical activity, quality of life, and sleep quality in heart failure (HF) patients, and further examined the difference by age and sex. The authors found lower levels of total or walking PA during lockdown both in younger and older patients. The paper wrote well, however, I have some comments as below.

  1. The greatest strength is that this paper added new data about the effects of the COVI-19 lockdown on HF patients. However, the biggest problem is whether it supplied the sufficient novelty. The authors should better clarify the novelty, significance in the abstract and introduction.

 Thank you very much for your comment. Your comment has been addressed. “The evidence about HF patients is scarce, and, to the best of our knowledge, no literature makes a differentiation based on sex and age in HF patients in this regard so this research is novel in this field”

  1. The another strength of this study is to examine the difference with specific comparisons by age and sex, as a results, the authors should added the results of clinical and demographic characteristics stratified by age and sex.  

 Thank you very much for your comment. Your comment has been addressed.

  1. Except for lockdown, I think some other issues such as the social economic status, and mental health may also influence on the above aspects, and the causal inference problem should be noticed.   

 Thank you very much for your comment. Your comment has been addressed.

  1. The full stopis not necessary after the title.

Thank you very much for your comment. Your comment has been addressed.

Reviewer 2 Report

Comments and Suggestions for Authors
introduction
The introduction explains the situation in Spain due to COVID-19, but COVID-19 was a worldwide problem. Please add information on the problems caused by COVID-19 in various regions other than Spain, such as Asia, Europe, and the Americas.
Outcomes and Measures
In the research method, the reliability and validity of the questionnaire were written only in writing, so please present the reliability and validity of the questionnaire in a table.
result
The contents of the results are properly structured.
discussion
In the discussion, comparisons with existing studies were properly structured. However, the reason why the hypotheses were rejected should be supplemented by comparing them with previous studies

Proper research design was made and the results of the study were meaningful. However, there is some lack of supplementary explanation for some rejected hypotheses. I'd like you to supplement the rejected hypothesis a little more. Thank you for submitting a good paper.

Comments on the Quality of English Language

Minor editing of English language required

Author Response

Introduction

The introduction explains the situation in Spain due to COVID-19, but COVID-19 was a worldwide problem. Please add information on the problems caused by COVID-19 in various regions other than Spain, such as Asia, Europe, and the Americas.

Thank you very much for your comment. Your comment has been addressed.

Outcomes and Measures

In the research method, the reliability and validity of the questionnaire were written only in writing, so please present the reliability and validity of the questionnaire in a table.

Thank you very much for your comment. Your comment has been addressed.

Result

The contents of the results are properly structured.

Thank you very much for your comment.

Discussion

In the discussion, comparisons with existing studies were properly structured. However, the reason why the hypotheses were rejected should be supplemented by comparing them with previous studies. Proper research design was made and the results of the study were meaningful. However, there is some lack of supplementary explanation for some rejected hypotheses. I'd like you to supplement the rejected hypothesis a little more. Thank you for submitting a good paper.

 Thank you very much for your comment. Your comment has been addressed.

Round 2

Reviewer 1 Report

Comments and Suggestions for Authors

The authors have addressed my concerns about this paper. I have no more comments.